Published in FAST Workshop on Smalltalk Related Technologies (11/2022)

# On Possible Extensions to the Smalltalk Syntax

**Leandro Caniglia**
*Fundación Argentina de Smalltalk*

*leandro.caniglia@fast.org.ar*

**Javier Burroni**
*University of Massachusetts, Amherst*

*javier.burroni@gmail.com*

**Reviewed on OpenReview:** *https://openreview.net/forum?id=oPQBinIjBt*

## Abstract

Even though changing it is fairly easy, the `Smalltalk` syntax has remained essentially stable for many decades. The reason is that people are cautious about side effects that could mess up something provably good. This doesn't mean that there aren't variations one might propose, but that the real task is to identify a coherent, non-distorting, set of extensions that increase the expressiveness of the language in favor of new possibilities related to areas such as interoperability, formal reasoning and mathematics.

## 1 Introduction

Modifications to the traditional `Smalltalk` syntax not only involve touching the `scanner` and the `parser`, they also require adapting the tools and, sometimes, adding or modifying base classes or even the `vm`. For instance, Unicode support [cf. Berman (2022), Vuletich et al. (2022)] for `Smalltalk` (as for any other system) not only requires going beyond `byte strings`, it also calls for convenient ways of typing characters that aren't present in the keyboard. And it is not just that. As soon as we allow for, say, Unicode selectors, we will be asking ourselves new questions. Wouldn't it be nice if we were able to define[1]

```
· aNumber
    ^self * aNumber
```
and
```
Number ≫ ²
    ^self · self
```

Because then, we would likely want to allow subscripted variables as in

```
x: t
    ^x_0 + (v_0 · t) + (a / 2 · t ²)
```

etc., in a race whose end is not easy to anticipate.

And this is not just about mathematics. What if we wanted to inject a `Json` literal such as

```
json := {
    "x" : 3,
    "y" : 4
}
```

or had the aspiration to promote the current `IR`, which displays bytecodes in human readable form, into a low-level (sub)language of `Smalltalk`? It could be useful to bring primitive methods into the image such as

```
SmallInteger ≫ highBit
    <IL>
    load R with high bit
```

---

[1]Methods and code snippets in LaTeX have been automatically generated from `Smalltalk`.

for the sake of performance or expressiveness, for testing new ideas without having to go through long processes involving other tools.

In what follows we will discuss these kinds of questions and will consider some possible extensions to the `Smalltalk` syntax, all of which have been used for years by a group of professional software makers. The prevalent idea behind these extensions is aligned with the need to let `Smalltalk` include features that would otherwise belong in external files or tools that affect the *configuration management* of complex systems, something we want to avoid as much as possible.

Our main contribution is a new method for the harmonic integration of *foreign* code into `Smalltalk` [Sections §4.1 and §4.2]. In contrast to the existing use of `Strings` for the representation of foreign structures, this method will check the syntax we intend to conform to and let us know of any defect at compilation time [Sections §5.1 & §7.8]. The foreign code will benefit from formatting and pretty-printing, which wouldn't be accessible using the `String` representation [Section §7.8]. In addition, we propose the extension of the `Smalltalk` syntax to allow for *prefixed unary* selectors [Section §3.2] as well as *unary symbols* [Section §3.3] and add flexibility to curly braces so that they can also be used to represent `Json` objects [Section §3.1]. Finally, we will illustrate the usefulness of these capabilities when applied to self-hosted runtimes [Section §5.3] and symbolic solvers implemented by the `Z3` library [Section §7].

## 2 Compilation

Before going into further details, let's sketch the compilation process that we are assuming:[2]

1. The `scanner` consumes source code characters and produces tokens.
2. The `parser` consumes tokens and produces nodes.
3. The `semantic analyzer` visits the node-binding variables.
4. The `generator` consumes nodes and produces bytecodes.
5. The `compiler` builds the method with all the information obtained above.
6. The `nativizer` consumes bytecodes and produces machine code.

## 3 Low Hanging Fruits

In this section we introduce some extensions to the `Smalltalk` syntax, which despite being (relatively) easy to implement, effectively expand the expressiveness of the traditional *unary-binary-keyword* triad.

### 3.1 The `Json` trick

To get started with the spirit of our work, let's consider a familiar code snippet example such as

```
json := {'x' -> 3. 'y' -> 4} asDictionary
```

which we might honestly consider acceptable, but hardly elegant. A better possibility would have been

```
json := {
    'x' : 3,
    'y' : 4
}
```

The interesting remark is that this is *valid* syntax on any `Smalltalk` dialect, whose evaluation would create an `Array` with only one element given by the meaningless (compound) message

```
(('x' : 3) , 'y') : 4
```

---

[2]There are also `optimizers` involved but they aren't relevant for the discussions that follow.

Can we alter this interpretation into something useful for our purposes?

Firstly, we need to make sure that `colon` is a valid selector. While this is not `ANSI Smalltalk`, enabling this extension is not a problem (as some dialects have already done [cf. Vuletich et al. (2022)]). Therefore, we can assume that the expression is correctly formed. Hence, we only need to tweak the standard meaning of the expression above so as to produce the intended `Json` object we are trying to express. Undoubtedly, we want to accomplish this without sacrificing the traditional notation for creating arrays with curly braces. But this is doable. After all, what's inside the curly braces is easily identifiable: a sequence of binary messages whose selectors alternate between `colon` and `comma`, where the receiver is either a literal string or a message of the same kind.

Since the curly brace notation already forces the parser to do something special, we could in fact tweak such behavior and let the parser decide whether the content of the BraceNode isn't better interpreted as a `Json` object. Otherwise, if what's inside the curly braces doesn't conform to the `Json` pattern, we would obtain the expected Array. In other words, we need a way to decide what's inside a BraceNode. Something on the lines of

```
hasJsonFormat
    ^JsonAssociationVisitor new visit: self
```

which will let the compiler decide whether it should interpret the BraceNode as an Array or a JsonObject. We implemented the parse tree visitor subclass and it only required writing visit methods for Block, Literal, BraceNode, Identifier and Message, where the last one is in charge of checking for the alternating pattern. More importantly, our experience showed that the trick wasn't a bad idea since its benefits outweighed the affordable cost of implementing it; especially because it supports the use of variables and unary messages in the values of the associations, and does it for free. Here is a real-life example:

```
addMember: aMember as: aRole
    ^self requester
        post: 'team'
        withJson: {
            'member' : aMember id,
            'role' : aRole id
        }
```

Note also that the nesting of BraceNodes perfectly aligns with the nesting required by `Json` structures.

This trick, however, is open to a valid critique, which is its lack of generality. In fact, the trick is specific to the `Json` syntax and depends on its contingent proximity to the `Smalltalk` language. We will address both issues below. But before that let's see what other simple things we can do to expand the `Smalltalk` syntax in useful, and natural, ways.

## 3.2   Prefixed Unary Selectors

Imagine you are teaching Boolean propositions and need to write the following method:

```
deMorgan
    ^p not & q not == (p | q) not
```

The question is, what has to be done to enable the implementation that best resonates with your intention:

```
deMorgan
    ^¬ p ∧ ¬ q ≡ ¬ (p ∨ q)
```

There are two issues here: (1) the Unicode characters $\neg$, $\wedge$, $\equiv$, $\vee$ and (2) the fact that $\neg$ should act as a *prefixed unary selector.*[3]

---

[3]Even though Unicode wasn't defined with Mathematics in mind, other useful prefixed selectors are: $\partial$, $\int$, $\sum$, $\forall$, $\exists$ etc.

A convenient (and proved) way to input Unicode characters is to adopt the LaTeX notation: let the programmer write the corresponding command, v.g., `\equiv␣`. Then, as soon as they press the spacebar (or any punctuation mark for that matter), the command changes to the intended Unicode character, namely $\neg$, $\wedge$, $\equiv$, $\vee$ etc.

The second question requires tweaking the SmalltalkParser. Note that the scanner will read '$\neg$' much as it does with any binary token. Therefore, the production we have to modify is the primary one, as it is there that the receiving expression is built. While this is dialect-dependent, our method should change on the lines of

```
primary

    token isNameToken ifTrue: [^self step asIdentifier].
    token isLiteral ifTrue: [^self step].
    (token is: $[) ifTrue: [^self block].
    (token is: $() ifTrue: [^self parenthesizedExpression].
    (token is: #'#(') ifTrue: [^self literalArray].
    (token is: ${) ifTrue: [^self bracedArray].
    (token is: #'#[') ifTrue: [^self literalByteArray].
    (token is: #'-') ifTrue: [^self negativeNumber].
    ^nil
```

```
primary

    • (compiler isPrefixSelector: token)
          ifTrue: [^self prefixedPrimary].
    token isNameToken ifTrue: [^self step asIdentifier].
    token isLiteral ifTrue: [^self step].
    (token is: $[) ifTrue: [^self block].
    (token is: $() ifTrue: [^self parenthesizedExpression].
    (token is: #'#(') ifTrue: [^self literalArray].
    (token is: ${) ifTrue: [^self bracedArray].
    (token is: #'#[') ifTrue: [^self literalByteArray].
    (token is: #'-') ifTrue: [^self negativeNumber].
    ^nil
```

Note the role played by the Compiler who will decide whether our token can be considered as a prefixed unary selector. This allows us to control which classes will configure the compiler to support prefixed unary selectors:

```
configureCompiler: aCompiler
    aCompiler prefixedSelectors: '¬'
```

Let's remark here that the change doesn't require any modification in the `vm` because the parser will create a regular method with unary selector $\neg$.

### 3.3   Unary Symbols

An example Smalltalkers usually give when first introducing message sends to others consists in debugging factorial:

```
Integer ≫ factorial
    self > 1 ifTrue: [^(self - 1) factorial * self].
    self < 0 ifTrue: [^self error: 'not valid for negative numbers'].
    ^1
```

Well, wouldn't it be more natural if we were allowed to write, say 5! rather than 5 factorial? Of course, we would like to enable this possibility without changing the `Smalltalk` grammar. In other words, we want to give classes the freedom to decide if a given binary symbol will represent a binary or a unary selector.

For instance, we could use the very same symbol '!' to signal Exceptions and even have both versions, unary and binary, coexisting in the same method:

```
Number ≫ !
    ^self > 1
        ifTrue: [(self - 1) ! * self]
        ifFalse: [self ≥ 0
            ifTrue: [1]
            ifFalse: [Error ! 'not valid for negative numbers']]
```

where we have used the Unicode characters '$*$' (typed `\ast␣`) and '$\geq$', (`\ge␣`). For another typical example consider

```
Number ≫ %
    ^self * 0.01
```

which allows us to write

```
vat
    ^self price * 10 %
```

The key observation is that there is no ambiguity in the sender, which in `Smalltalk` *decides the arity* of every message. For instance, an expression such as

```
5 ! + 2 %
```

presents no vagueness: both '!' and '%' act as unary messages while '$+$' acts as binary, *even though the three symbols are binary.* In other words, the parser can easily decide when a binary symbol corresponds to a binary or a unary message. Basically, this happens when the binary token ends the expression, i.e., it is the last one, it is followed by one of #($. $; $] $) $}) etc.[4]

With this syntax, it is the sender who knows whether the symbol represents a unary or a binary selector. Thus, sometimes parentheses must be used to enforce the desired interpretation. For instance, if we define

```
Number ≫ º
    ^self degreesToRadians
```

then the expression 180 º is valid, and we must write (180 º) sin to prevent the message from being interpreted as binary with argument sin.[5]

Note however that this change in the parser may require a change in the bytecode generator. The reason is that binary symbols might represent unary messages in some cases and binary in others (this is the proposal we stated). Therefore, if the vm assumes that every binary selector has arity 1, then the parser should push a dummy argument such as nil to compensate for the imbalance caused on the stack.

No additional push bytecode would be necessary if the vm computed the arity as 1 when the message is a binary send or as 0 in the case of regular sends. Note, by the way, how not having direct access to the vm prevents simple tweaks from being implemented. The very fact of feeling compelled to clarify whether the implementation of a new idea involves modifying the vm or not should be considered a sign of attention: why the distinction? hadn't we agreed that `Smalltalk` is utterly expressive? This is why self-hosted runtimes are better suited for innovation. We will get back to this issue later.

A similar side effect happens with the DNU mechanism. By the time the vm has to reify the message not understood it will need to decide whether the send was binary or unary as we just indicated. Note however that this can be remedied in the vi with the addition of a fixArguments message that will analyze thisContext to decided the arity of the binary selector:

```
doesNotUnderstand: aMessage
    aMessage fixArguments.
    ^MessageNotUnderstood message: aMessage receiver: self
```

Again, none of this is needed when the runtime is self-hosted.

**Digression.** As we have seen and will further illustrate in §7, the ability to combine Unicode characters with prefixed unary selectors and unary symbols allows the programmer to express in `Smalltalk` language used in Mathematics and Logic as well as in other disciplines such as Physics etc.

---

[4] As usual, especial care might be necessary when the next token is a negative number.
[5] Other examples: A* for A adjoint once the matrix multiplication selector is '·', A$^\mathrm{T}$ for A transpose etc.

### 3.4   Functional Syntax

The missing piece to better support mathematical formulas stems from the inability of `ANSI Smalltalk` to parse functional expressions like f(x), where both f and x are variables denoting objects with f being *evaluable*, i.e., responsive to the value: message.

Fortunately, we can take advantage of the fact that the `Smalltalk` grammar doesn't include any production for this kind of syntax [cf. § 4.3]. Once again, with no change in the scanner, the parser can be enhanced to cope with the new situation. All we need to do is include this possibility as a new kind of message:

```
expression
    | primary expression |
    (token isNameToken and: [self peek isAssignment]) ifTrue: [^self assignment].
    primary := self primary ifNil: [^self missingExpression].
  • expression := self function: primary.
    expression := self unarySequence: expression.
    expression := self binarySequence: expression.
    expression := self keywordSequence: expression.
    expression == primary
        ifFalse: [expression := self cascadeSequence: expression].
    token endsExpression ifFalse: [self errorIn: primary].
    ^expression
```

where

```
function: aParseNode
    (token is: $() ifFalse: [^aParseNode].
    ^compiler functionNode
        position: aParseNode position;
        receiver: aParseNode;
        arguments: {self parenthesizedExpression};
        end: token position
```

which requires the addition of `FunctionNode`, a new subclass of `MessageNode` whose hidden selector is value:. Note that the spirit of `Smalltalk`, which consists in making the most of message sends, is preserved: in f(x) both variables denote objects and the semantics is exactly f value: x. Only the notation changes to enhance the natural expression of mathematical formulas.

For instance, should the user promote the sin message to a function object, they could write $\sin(30°)$ rather than sin value: 30 degressToRadians, which requires a mental parsing and translation to reveal its actual meaning. Combining this syntax with `Unicode` selectors, we can express the inverse function $f^{-1}(x)$, where the unary message '$^{-1}$' stands for the inverse of f. Note that the feature is not restricted to variables; also valid are $(f+g)(x)$, $f \circ g(x)$, $f(x+y)$ as well as $f(A)$ and $f^{-1}(B)$ for sets.[6]

## 4   From Tricks to Capabilities

For some people the expressiveness of `Smalltalk` is so extraordinary that they would see no point in supporting the compilation of a foreign language. Why would someone choose to do that given the elegance of its unique syntax? We concur with that opinion and don't think that sometimes a different language would have provided better results. At the same time, we also think that there are other interpretations for this question which indicate that foreign code may still have a useful role to play in terms of expressiveness.

Even if we were to focus exclusively on `Json` (we aren't), what if we wanted to express a more complex `Json` object? What if its definition were readily available on the web for us to copy and paste? Well, we would need to replace double with single quotes before saving the expression in a method, because our trick cannot digest double quotes.

---

[6]Surprisingly enough [cf. Gamma function] with this capability $\Gamma(n) = (n-1)!$ becomes a valid `Smalltalk` expression.

Wouldn't it be nicer if we were allowed to compile our `Smalltalk` method with the structure copied as it is, without further editing?

Such a situation may occur when dealing with `Json` schemas, as is the case with the following method. Here the <json> pragma indicates that the code that follows obeys a specific syntax:

```
def
    <json>
    {
        "type": "object",
        "properties": {
            "firstBatch": {
                "type": "array",
                "items": {"$ref": "MongoDocument"}
            },
            "nextBatch": {
                "type": "array",
                "items": {"$ref": "MongoDocument"}
            },
            "partialResultsReturned": {"type": "boolean"},
            "ns": {"type": "string"},
            "id": {"type": "integer"}
        }
    }
```

What this method really does when executed is a different decision. It might answer with the `Json` string or the associated `Json` dictionary—a circumstance that the programmer can easily resolve by querying (or inspecting) the `CompiledMethod`. In either case, the method would bring the benefit of parsing, formatting and pretty-printing its *foreign* section. So, let's focus on these relevant aspects.

The parsing part is relevant because, as soon as we try to accept (save) the method, it will check the syntax we intend to conform to. This will readily let us know of any defect at compilation time. Formatting and coloring are also nice: had we had a literal `String` instead, the method would have looked meager, less revealing and even exotic. The visual connection with its meaning would have poorly resembled the `Json` format. This is of uttermost importance in the case of `Smalltalk`, whose practitioners are used to dealing with objects, not text.

So, what's our proposal? Is it about having or not a `Json` parser? Of course not. The proposal is not even about combining `Smalltalk` and `Json` in the same method. The possibility we want to consider seriously is why and how we should combine `Smalltalk` with *any* foreign source code, or any `DSL` for that matter.

## 4.1   Scanning

Following the style of `<pragma>` notations, one naturally arrives at *tagged nodes*, i.e., sections of the source code enclosed between tags, say `<pas>···</pas>` for `Pascal`, `<js>···</js>` for `Javascript` etc. More specifically, let's ask ourselves how the following code should be parsed:

```
error
    | pi json |
  • pi := <json>
        {
            "x": 355,
            "y": 113.0
        }
    </json>.
    json := JsonObject fromString: pi.
    ^json x / json y - Number pi
```

Well, when the SmalltalkScanner encounters the character '<' that comes after pi := it will create a binary token in the same way it would do when this character acts as a binary selector. At this point, this very token will be examined by the SmalltalkParser in an attempt to build a primary node for the expression that should follow the assign. Thus, the new syntax has no impact on the SmalltalkScanner, which may remain unchanged. As we will see below, the scanner will have a role to play though.

**Remark.** As an implementation detail, in the case where the method consists of a single tag, say <json>, followed by the entire body of foreign language [cf. §4], instead of considering the tag as a proper pragma [cf. Ducasse et al. (2016)] one might assume at parsing time that the closing tag is implicit.

## 4.2 Parsing

As pointed out above, by the time the SmalltalkParser reaches the first assign token := it will try to detect the corresponding right-hand-side expression. Well, it is at this point where an interesting condition takes place. Since the token it encounters is the symbol <, which should not happen here in the traditional syntax, under normal circumstances we would get a CompilerError. This gives us the chance to modify that behavior. In other words, the fact that < cannot follow the assign symbol is the hook we needed to chain the new syntax. Let's see how.

Firstly, we need to introduce a new subclass of LiteralNode, namely TaggedNode. While the particular details of this addition depend on the Smalltalk dialect, a few observations would clarify the idea and show its feasibility.

Secondly, this new node will need to keep two pieces of information, the tag and the string surrounded by the tag/untag pair, an ivar of TaggedNode we will call value.

Thus, a service the SmalltalkScanner can provide is the production of the nextTaggedNode, which will answer with an instance of TaggedNode. Note however that it is up to the SmalltalkParser to decide when the nextTaggedNode should be produced, because it is the one who knows when the next primary node is expected to come. While this is dialect-dependent, the following implementation may serve the purpose of conveying the idea:

```
primary
    token isNameToken ifTrue: [^self step asIdentifier].
    token isLiteral ifTrue: [^self step].
    (token is: $[) ifTrue: [^self block].
    (token is: $() ifTrue: [^self parenthesizedExpression].
    (token is: #'#(') ifTrue: [^self literalArray].
    (token is: ${) ifTrue: [^self bracedArray].
    (token is: #'#[') ifTrue: [^self literalByteArray].
    (token is: #'-') ifTrue: [^self negativeNumber].
•   (token is: #'<') ifTrue: [^self taggedNode].
    ^nil
```

The method taggedNode only needs to send scanner nextTaggedNode and then advance (a.k.a. step). As usual, every potential parsing error (in this case the otherwise upcoming `missing primary`) offers an opportunity to extend the syntax of the programming language.

This is all we need in order to incorporate tagged nodes into our repertoire of parse nodes. Note the generality of the approach: it is not restricted to any particular tag/untag pair. Of course, so far our TaggedNode is only able to capture the String value enclosed between tags, which is already useful, but not quite. Hence, some more work is in order.

**Remarks**

i) Our extension to the primary production enables tagged nodes to occur not only in assignments but also as message arguments and return expressions (there are examples in the sections below).

ii) The `def` method given in §4 illustrates how the SmalltalkParser generates an *implicit* ReturnNode whose expression is the entire foreign code. As a result, the programmer is freed from the need to explicitly add the return symbol `^` and closing the `<json>` tag, which would have overloaded the body of the method.

iii) Our implementation of `<tag>···</tag>` pairs turns the context-free grammar of `Smalltalk` into a context-sensitive one. The main side effects of this change are (a) tagged nodes cannot be nested, and (b) the substring `</tag>` cannot occur inside the tagged node.[7] One way to keep the grammar context-free is to use `</>` as the closing tag instead.[8]

## 4.3 Grammar Holes

In order to be able to extend the `Smalltalk` grammar to allow for Prefixed Unary Selectors and TaggedNodes, we need to make sure that we still allow to write everything that we were able to before the extension. A way to achieve this is by using *Grammar Holes*, i.e., combinations of words and symbols that the current grammar does not support and produce CompilerErrors. Both extensions presented rely on modifying the behavior when a binary selector is encountered. In the original grammar (Goldberg & Robson, 1983), these selectors appear only in two rules,

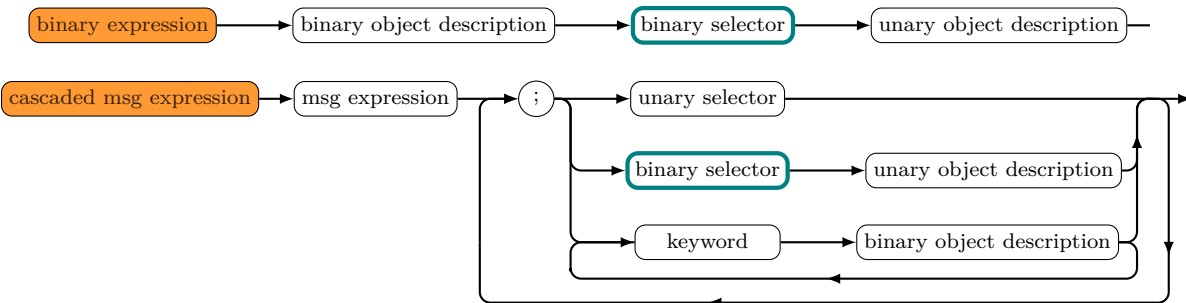

The semantic interpretation of these two rules is clear: both the left- and right-hand sides should be parsed to objects. As long as we preserve this property, we are allowed to extend the `Smalltalk` grammar. And that is exactly what we do in Sections §3.3 and §4.2.

Prefixed Unary Selectors were implemented by ending the expression just after the binary selector, i.e., without the unary object description, and similarly TaggedNode was implemented by considering a binary selector placed without a preceding binary object description (or ';').

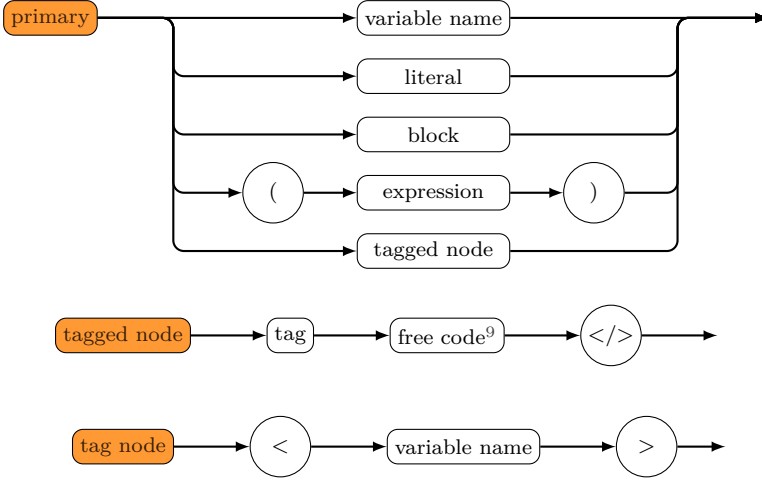

---

[7]For instance, `</xml>` would become forbidden in the `XML` body.
[8]This distinction becomes irrelevant for languages like `Assembly` where, say, `</asm>` doesn't happen.
[9]A free code is any combination of characters excluding `</>`.

## 5   Foreign Nodes

One piece of information that is needed to map tags with foreign languages is easily achieved by means of a *registry* known to the Compiler. Every foreign parser (or compiler) that we happen to support will be available to programmers as soon as the association tag → parser is entered into the registry. If a given tag has been registered, a ForeignNode will be created with its parser ivar initialized to the corresponding DSL parser. This is a new kind of ParseNode whose responsibility is to compile the value of the associated TaggedNode.

```
TaggedNode ≫ body: aString
•   value := self compiler nodeForTag: tag.
    value value: aString.
•   value ast isNil ifTrue: [value := self compiler stringNode value: aString].
    value
        position: stretch start + tag size;
        end: stretch start + tag size + aString size - 1
```

The ForeignNode is created in the first line. The parsing of the foreign code happens in the second, i.e., as a result of sending the message value:.

If the tag is not registered, we simply create an agnostic StringNode. We could have signaled a Compilation-Error instead, but the default behavior chosen acts as a placeholder calling for an appropriate resolution that the programmer may have decided to defer, which is more intention revealing than any comment that might be attached to the String. This also happens if the DSL parser fails to produce an abstract syntax tree from the string provided. It is at this point where the Smalltalk programmer will know that the foreign code doesn't conform to the specified syntax.

```
ForeignNode ≫ value: aString
    ast := parser parse: aString
```

### 5.1   Other Target Languages

Let's see a few examples of the new capability in contexts where we've actually operated with them. Some rely on PetitParser [cf. Kurs et al. (2013)], others in *ad hoc* parsers suitable for the foreign language.

There are also examples for C, Intel 64/32, XML, CSS, Html, and others. In all cases, foreign languages can occupy the entire body of the method, provided the appropriate pragma, or they can be injected as literals enclosed in TaggedNodes.

**Javascript.** This language is so ubiquitous that its inclusion doesn't require much justification. We use it to support the rendering part of web applications served by Smalltalk.

```
passwordError
    <js>
    $scope.passwordError = function(form) {
        if (!shouldValidate(form, "password"))
            return null;
        if (form['password'].$error.required)
            return "The password is required";
        return null
    }
```

**Pascal.** Although not very popular these days, Pascal is used by Inno Setup, the free installer for Windows applications.

```
iniSectionsFromInf
    <pas>
    procedure iniSectionsFromInf(sections: TStrings);
        var
            all: TStringList;
            i: Integer;
            section: String;
        begin
            all := TStringList.Create;
            sectionsFromInf(all);
            for i := 0 to all.Count - 1 do
                begin
                    section := all.Strings[i];
                    if isIniSection(section) then sections.Append(section);
                end;
            all.Free
        end;
```

## 5.2   Dynamic Passing of Arguments

In our experience, there are cases where *parametric* foreign code simplifies the generation of the desired output:

```
varsDeclaration: url
version: version
product: product
client: client
    <js>
    $scope.baseUri = #url;
    $scope.selectedRelease = #version;
    $scope.selectedProduct = #product;
    $scope.selectedCompany = #client;
```

The example illustrates the declaration of `Javascript` variables, where the same foreign method can be used for the production of foreign code based on the formal arguments provided by the `Smalltalk` signature.

Note that in the body of the method its arguments are distinguished by means of the pound prefix '`#`'. In this case the method uses an instance of ParametricString, which models the body of the foreign code annotated with occurrences of symbolic arguments. These arguments are replaced dynamically every time the method is invoked. This possibility prevents the proliferation of methods that only differ in parameters easily obtained at runtime.

**Remark.** The support of method arguments as parameters to foreign code is general. What's not general is the choice of a universal prefix or delimiter because that is language-dependent. Thus, it is up to the programmer to decide which one to use (taking advantage of grammar holes in the target language). For instance, in `C` we used `$` as a prefix. In our implementation this decision can be materialized in the method configureCompiler: by setting up the parametricDelimiters option [cf. § 5.5]. Even though we have only enabled this capability for methods where the entire body is foreign, in § 5.4 we show another way to achieve the same in the case of foreign code injection.

## 5.3   Assembly

The use of `Assembly` in `Smalltalk` methods deserves special attention. In systems that have Dynamic (or Live) Metacircular Runtimes, where the `vm` is self-hosted, i.e., implemented inside the very same `vi` it runs, there is the need to produce `Assembly` code on the fly from within the `vi`. This process, called *nativization*, dynamically compiles (target) `machine code` from bytecodes. And since this requires the implementation of a corresponding InstructionEncoder, some more work along the same ideas results in its reciprocal `InstructionDecoder`.

Given the limited number of bytecodes, fragments of the machine code generated repeat a lot. For that reason it makes no sense to go through the complete nativization process: it is enough to use `Assembly` *templates*, i.e., pre-compiled versions of meaningful fragments of the required machine code. For instance,

```
saveCallerFrame
    self assemble: <x64>
        000:   push rbp
        001:   mov rbp, rsp
    </>
```

injects the literal #[85 72 137 229] as the argument of `assemble:`, as revealed by the IL:

```
#saveCallerFrame
    push literal #[85 72 137 229]
    send selector #assemble:
    return self
```

Of course, the alternative would have been to provide the `ByteArray` instead. However, compare the expressiveness of

```
convertAtoSmallInteger
    self assemble: #[72 209 226 72 255 194]
```

with

```
convertAtoSmallInteger
    self assemble: <x64>
        000:   sal rdx, 1
        003:   inc rdx
    </>
```

Without this capability the programmer is forced to resort to an external tool to find each of the intended `ByteArrays`. More work for the writer, less clarity for the reader; a silly loss-loss situation. Note also that targeting `Intel 32` instead only entails replacing the tag x64 with x86. The simple comparison between both versions of the method addresses the question of why we should take the trouble to inject foreign code in `Smalltalk`: *to make the most of its expressiveness.* As we will see next, there are still more reasons.

### 5.4   Parametric Assembly

In some cases the dynamic injection of parameters is useful for producing `machine code` templates:

```
loadRwithAindex: index
    | offset |
•       index = 1 ifTrue: [^self assemble: <x64>mov rax, qword ptr [rdx]</>].
    offset := index - 1 * wordSize.
    (-128 <= offset and: [offset < 128])
        ifTrue: [^self
•           assemble: <x64>mov rax, qword ptr [rdx + imm8]</>
            imm8: offset].
    (-0x80000000 <= offset and: [offset < 0x80000000])
        ifTrue: [^self
•           assemble: <x64>mov rax, qword ptr [rdx + imm32]</>
            imm32: offset].
```

where we use the symbolic parameters imm8 and imm32 to let the template inject the actual values at runtime, when offset is realized.

It is pertinent here to note the clarity contributed by the `Assembly` code and the pretty-print of the method that conveys meaning to otherwise cryptic ByteArrays. These properties encourage the implementation of more precise logic that takes advantage of the possibilities offered by the target `Assembly`. When there is no need of any immediate displacement, the shortest version of the instruction is used, and the same goes for the other two bit-lengths of signed `byte` and `double word`. Ponder also the pedagogical value of programming/browsing with this kind of aid.

Another aspect to underline has to do with ease of reproduction. Should the user need to program a similar template for a different pair of registers, the modification would have required looking at an external tool for the new bytes, a task that is boring and error prone. Here, in contrast, those changes are trivial and immediate, and are also verified by the underlying parser. For instance, to implement loadEwithFPindex: the programmer would have only to replace rax with rsi and rdx with rbp. Of course, registers could have been parametric too, but the conclusion remains the same.

### 5.5    Intermediate Language

Special mention deserves the support for adding a low-level language that uses the syntax of the `Intermediate Representation` (a.k.a. `IR`) of bytecodes. The same method we saw in § 3.2, that classes implement when they need to set specific options to the compiler, can be used here to override the default SmalltalkCompiler. For instance,

```
configureCompiler: aCompiler
    aCompiler optionAt: #compilerClass put: IntermediateLanguageCompiler
```

allows every class to choose the actual compiler that will be used for its methods. The default class is SmalltalkCompiler (every dialect uses a different name for it) and the example shows how to modify this value. In this case, IntermediateLanguageCompiler allows the programmer to *skip* the step of using the `Smalltalk` syntax to produce a CompiledMethod.

Most `Smalltalk` dialects make use of bytecodes. The examples that follow correspond to the case where bytecodes make use of *registers* [cf. Wirfs-Brock & Caudill (1999)]. Since bytecodes are instances of ByteArray, a human-readable version is usually available for inspection. For instance, the bytecodes of the method above may look on the lines of

```
configureCompiler: aCompiler
1   load R with argument aCompiler
2   push literal #compilerClass
3   push assoc #IntermediateLanguageCompiler -> IntermediateLanguageCompiler
4   send selector #optionAt:put:
5   return self
```

The interesting thing to notice here is that we can use our capability for injecting foreign code to promote this intermediate *representation* into an actual intermediate *language* or `IL`. In fact, the very same parser used for displaying `IR` can be used for programming new methods. For instance, the method above could as well have been written as

```
configureCompiler: aCompiler
    <IL>
    load R with argument aCompiler
    push literal #compilerClass
    push assoc IntermediateLanguageCompiler
    send selector #optionAt:put:
    return self
```

producing a CompiledMethod with the corresponding bytecodes and whose literal frame consists of the global association for the class IntermediateLanguageCompiler and the symbols compilerClass and optionAt:put:.

Of course, this is an artificial example, because the method is much better expressed in `Smalltalk` code. However, every dialect includes primitives whose logic cannot be expressed in `Smalltalk`, or which might but incurring in a significant performance penalty or some other inconvenience.

An example of this latter circumstance is the method SmallInteger ≫ highBit that returns the largest position of 1 occurring in the binary representation of the receiver:

```
SmallInteger ≫ highBit
    self < 256 ifTrue: [
    ^#[
        0 1 2 2 3 3 3 3 4 4 4 4 4 4 4 4
        5 5 5 5 5 5 5 5 5 5 5 5 5 5 5 5
        6 6 6 6 6 6 6 6 6 6 6 6 6 6 6 6
        6 6 6 6 6 6 6 6 6 6 6 6 6 6 6 6
        7 7 7 7 7 7 7 7 7 7 7 7 7 7 7 7
        7 7 7 7 7 7 7 7 7 7 7 7 7 7 7 7
        7 7 7 7 7 7 7 7 7 7 7 7 7 7 7 7
        7 7 7 7 7 7 7 7 7 7 7 7 7 7 7 7
        8 8 8 8 8 8 8 8 8 8 8 8 8 8 8 8
        8 8 8 8 8 8 8 8 8 8 8 8 8 8 8 8
        8 8 8 8 8 8 8 8 8 8 8 8 8 8 8 8
        8 8 8 8 8 8 8 8 8 8 8 8 8 8 8 8
        8 8 8 8 8 8 8 8 8 8 8 8 8 8 8 8
        8 8 8 8 8 8 8 8 8 8 8 8 8 8 8 8
        8 8 8 8 8 8 8 8 8 8 8 8 8 8 8 8
        8 8 8 8 8 8 8 8 8 8 8 8 8 8 8 8]
            at: self + 1].
    self < 512 ifTrue: [^9].
    ^self highDigit highBit + (self sizeInBytes - 1 * 8)
```

Compare with the `IL` version of the same method:

```
SmallInteger ≫ highBit
    <IL>
    load R with high bit
```

which is shorter and inexpensive, as shown by its `Intel 64` assembly translation

```
SmallInteger ≫ #highBit
●  000:  bsr rax, rax                    ; compute highBit
   004:  sal rax, 0x1                    ; tag as integer
   007:  inc rax
   00A:  mov rsi, qword ptr [rbp - 0x8]  ; restore caller frame
   00E:  mov rbx, qword ptr [rbp - 0x10]
   012:  ret
```

It is worth remarking that the same capability can be used to access object header fields, namely hash, size, flags and behavior (or class, as is the case in most dialects).

The intermediate language also allows us to access Process fields such as framePointer and stackPointer, which are needed for stack manipulation, debugging and continuation support. These are indications of the benefits of self-hosted runtimes, where new bytecodes can be added following a relatively simple procedure and become immediately available for testing and further experimentation.

## 6    Consequences

This work is antipodal to replacing `Smalltalk` with other languages. It has to do with embracing `Smalltalk` and taking it to new territories that would otherwise have imposed external files and tools, cluttering the

*configuration management* of the system [cf. SEI]. Having inclusive proximity in the `Smalltalk IDE` to every `DSL` we need to work with is paramount to the principle of *uniform metaphor* [cf. Ingalls (1981)]. It is not only a question of expression and representation but also of beauty and harmony in a endless effort aimed at removing sources of impedance.

REASONING. As discussed in the bibliography [cf. Cadar & Sen (2013)], literal injection techniques are better suited when they conform to the following criteria: *Responsibility, Expansion Typing, Context Dependence, Segmentation, Segment Typing* and *Capture*. It is easy to see that our proposal clearly meets them. For example, *Responsibility* follows from the fact that tags are declared in the `Compiler` registry [Section §5], *Expansion Typing* [Section §4], *Capture* from the capability to inject parameters [Sections §5.2 and §5.4] etc.

METAPROGRAMMING. The advantage of having objects rather than strings or bytes is that this enables the implementation of features by means of metaprogramming. One example is the automatic generation of `Smalltalk` code for classes representing `C` structures. Consider for instance `HTTP_RESPONSE_INFO`. This is represented by a `Smalltalk` class with the same name that implements (class side) its definition

```
def
    <C>
        typedef struct {
            HTTP_RESPONSE_INFO_TYPE Type; // 0
            ULONG Length; // 4
            PVOID pInfo; // 8
        } HTTP_RESPONSE_INFO, *PHTTP_RESPONSE_INFO; // sizeInBytes = 16
```

which the programmer pasted from some `.h` file. The system not only compiled and pretty-printed the foreign language, it also computed the offsets and annotated them as comments. Apart from that, it generated a local pool dictionary with the symbols `Type`, `Length`, `pInfo` and `sizeInBytes` associated to the offsets they represent, and used them to compile all getters and setters required by the external structure. For instance,

```
info                            pInfo                          Length
    ^String                         ^self addressAtOffset: pInfo    ^self uLongAtOffset: Length
        fromMemory: self pInfo
        length: self Length
```

show how the programmer takes advantage of automatically generated lower-level getters to produce their method of interest, namely `info`. This kind of metaprogramming gets rid of tedious work which would also have been error prone.

SYNERGY. While the ability to support *external* `DSL`s for the sake of knowledge capture, literal injection etc. has many applications, care must be taken not to load on the shoulders of the user additional tasks related to updating, maintenance and scalability. This is why it is so important to analyze what else `Smalltalk` has to offer to better integrate scientific languages as *internal* (a.k.a. *embedded*) `DSL`s by means of supporting logical and mathematical symbols, notations and formulas. General features aimed at integrating mathematical expressions with `Smalltalk` code will allow more users to build elaborated scientific models without the need to rely on any exogenous `DSL`, enforcing that way a sound strategy of long-term synergy.

# 7 Application to Symbolic Computation

While advanced tools for automatic reasoning that produce formal proofs and derive solutions to logical and algebraic problems have become critical to some areas of the software industry, their adoption as a part of development methodologies still has a long way to go. In the next few sections we present a case study that, taking advantage of the capabilities discussed above, sheds light on the significance of the paradigm shift that results from a more systematic appropriation of these technologies. Although the topic has interest in itself, we present it here as a way to validate the syntactic extensions discussed in the previous sections.

## 7.1 Scenario

Within the realm of `vm` development the authors have experience in metacircular ones, which are written in the very `IDE` they are destined to run. These `vms` model the entire computational space in user (i.e., programmer) code and connect to the 'bare metal' by compiling virtual objects (e.g., bytecodes) into native code, or *transpiling* it into an ubiquitous language such as `C` or `Javascript`. Of course, nativization should not have defects; otherwise the entire construction will collapse. This places this part of the `vm` in an ideal position to be studied with the help of formal (a.k.a. symbolic) verifiers.

## 7.2 Approach

The approach commonly recommended by experts in automatic solvers (v.g., `Z3`) consists in transforming sections of the source code into a format more palatable to these tools. This has two consequences: (1) the code to be analyzed is first simplified, and (2) the verification is deferred to a different computational space. These characteristics create spatial and temporal separations between the exercise of development and the execution of formal analysis, which go against the immediacy pursued by agile practices. Our idea, in contrast, is to incorporate a broad range of `Z3` [cf. Moura & Bjørner (2008)] capabilities into the `Smalltalk IDE`, exploring how far we can reach in the effortless and natural transition from concrete to symbolic execution.

## 7.3 The Problem

Regardless of the fact that it is completely expressed in the target language, the implementation of live metacircular `runtimes` recognizes the existence of three main abstraction layers: *user code*, *intermediate representation* (`IR`) and *native code.*

Transformations from one layer to the next are core `runtime` tasks and require careful testing. In consequence, they constitute excellent candidates for formal verification. This goal, while highly ambitious, is nevertheless the right one for producing a truly reliable `vm`. The question arises: where could we start?

From the two opportunities at our disposal: proving the validity of `IR` behavior and of the target native code, the latter works better as an experimental first step because it has the advantage of being more basic.

In sum, the problem we are about to address consists in providing, within the same `IDE`, a seamless way to let the `runtime` programmer express and prove *theorems* referred to the model of a real cpu.

## 7.4 Model

The definition of our problem requires modeling a `cpu`. The relevant classes are

    CPU
    Operand
    ProgramStream
    RAM

where `CPU` stands for an abstract class with subclasses specific to concrete architectures such as `X86`, `X64`, or virtually any other. The class `Operand` is abstract with concrete subclasses

    Operand

Mem
Reg
   Flags[10]

Immediate operands are held in existing classes such as Byte, ExternalInt16, ExternalInt32 etc.

### 7.5 Execution

The model allows the programmer to seamlessly debug the program in the virtual cpu, reasoning about the behavior of every component for the sake of testing and understanding. As an example, consider the following test. The crucial part happens when each of the three assembly instructions is executed. What's interesting to examine here is the program flow triggered by assemble: and step:.

```
testAdd
    cpu
        assemble: <x86>
                    000:    mov al, 0x71
                    002:    mov dl, 0xf
                    004:    add al, dl
        </>;
        step: 3.
    self
        assert: cpu al equals: 0x80;
        assert: cpu carry equals: 0;
        assert: cpu overflow equals: 1
```

Details about the implementation of these messages can be found in Appendixes A and A.1.

### 7.6 Symbolic Execution

Now let's reflect on how we can write unit tests for our model. The test testAdd introduced above serves the noble and desirable goal of providing coverage to the implementors of mov and add. In this regard, we would normally observe that additional coverage is needed for other registers, memory operands etc. But what about adding more numerical examples? Since there are two operands at play, we would need to cover the range of $256 \times 256$ possibilities. And what about 32- or 64-bit operands? Would the idea of testing all the possibilities cross our mind? Of course not. At best we would test border cases and then write some assertions for randomly generated operands.

This recognizable train of thought, that may look adequate for practical purposes, is actually a barrier to more rigorous approaches. It trades coverage for generality, sparse cases for abstractions, examples for formal proofs. An interesting question arises: can we do better?

Let's examine in more detail how we could use the Z3 library. Firstly we simply ask our model to useZ3. There are three classes needed to connect with Z3: CPU, Reg and RAM. Here is all we need to do:

```
CPU ≫ useZ3
    z3 := Z3.Context default.
    registers do: [:reg | reg useZ3].
    self initializeDecoder
```

```
RAM ≫ useZ3
    "ArraySort of BitVectorSort"
    contents := cpu z3 Memory: 'RAM' wordSize: cpu wordSize
```

---

[10]Although some architectures do not have a `flags` register, the class exists for the ones that do.

```
Reg ≫ useZ3
    contents := cpu z3 BitVecConst: name size: self wordSize * 8
```

After sending the message `cpu useZ3` all `cpu` registers and memory will become `Z3` objects polymorphic with their `Smalltalk` counterparts. Let's now see the significance of this change from the programmer's viewpoint. Firstly, the programmer may add this message to their tests. For instance,

```
testAdd
    cpu
        useZ3;
        assemble: <x86>
                000:   mov al, 0x71
                002:   mov dl, 0xf
                004:   add al, dl
        </>;
        step: 3.
    self
        assert: cpu al equals: 0x80;
        assert: cpu carry equals: 0;
        assert: cpu overflow equals: 1
```

## 7.7 Generality

Let's analyze the possibilities created by having connected our model to `Z3`. Since now the registers are `Z3` bit-vectors, there is no need to assign constant values to them such as `0x71` or `0x0f`. In other words, we are now entitled to predicate general assertions. After this realization a new horizon of possibilities opens up before us. Consider this fragment:

```
cpu
    useZ3;
    assemble: <x86>
            000:   mov bl, al
            002:   add al, cl
    </>;
    step: 2
```

It saves register `al` into `bl` and then replaces `al` with `al + cl`. Note the generality of the code: no particular value is involved.

Here it is important to understand that `Z3` does not exhaust all the possibilities. Instead, it reasons formally following the logic of our implementation. In particular, the same verification would happen had we tested with `64`-bit registers with `add rax, rcx`.

The next sections explore some of the possibilities enabled by this kind of symbolic testing.

### 7.7.1 Theorems

We can now consider what-if assertions such as

```
cpu cl ≡ 0  ⟹  (cpu overflow ≡ 0)
```

which become actual *theorems* for `Z3` to prove or disprove. To add more expressiveness to our framework we can extend the `assert:` family of methods as follows:

```
assert: aBoolean
    | boolean |
    boolean := aBoolean isBoolean
        ifTrue: [aBoolean]
        ifFalse: [aBoolean isTautology].
    super assert: boolean
```

```
isSatisfiable
    | solver result |
    solver := context makeSolver.
    result := solver assert: self; check.
    solver release.
    ^result == true
```

```
Z3.Bool ≫ isTautology
    ^(¬ self) isSatisfiable not
```

which expresses the fact that the expression has at least one model.

Equipped with this capability we can easily attach theorems to our test and assert their satisfiability:

```
testZ3Add
    | thm1 thm2 |
    cpu
        useZ3;
        assemble: <x86>
                000:   mov bl, al
                002:   add al, cl
        </>;
        step: 2.
    thm1 := cpu cl ≡ 0  ⟹  (cpu overflow ≡ 0).
    self assert: thm1.
    thm2 := (cpu bl < 0) ∧ (cpu cl < 0) ∧ (cpu al ≥ 0)  ⟹  (cpu overflow ≡ 1).
    self assert: thm2
```

Note that Boolean expressions such as the ones illustrated by the test constitute actual *theorems* because of their complete generality.[11] It is also worth remarking how Unicode selectors elegantly clarify the meaning of logical expressions in comparison with ANSI Smalltalk and C:

| Unicode Smalltalk | ANSI Smalltalk | C |
|---|---|---|
| ¬ p | p not | Z3_mk_not(c,p) |
| p ≡ q | p === q | Z3_mk_eq(c,p,q) |
| p ⟹ q | p ==> q | Z3_mk_implies(c,p,q) |

### 7.7.2   Remarks

We can take advantage of the same capability to add remarks such as

```
testZ3AddRem
    | rem1 rem2 |
    cpu
        useZ3;
        assemble: <x86>
                000:   mov bl, al
                002:   add al, cl
        </>;
        step: 2.
    rem1 := cpu carry ≡ 0.
    self deny: rem1 isTautology.
    rem2 := cpu carry ≡ 1.
    self deny: rem2 isTautology
```

which means that the carry flag will eventually take both values 0 and 1 after an addition.

---

[11]Even though our examples use registers, tests can predicate on memory operands too.

### 7.7.3 Counterexamples

We can also ask Z3 to find counterexamples to theorems that our assembly program doesn't imply. This is accomplished by asking a model from the negation of the invalid theorem.

```smalltalk
testZ3Add3
    | thm model cl bl al cy ov |
    cpu
        useZ3;
        assemble: <x86>
                000:    mov bl, al
                002:    mov cl, byte ptr [edi]
                004:    add al, cl
        </>;
        step: 3.
    "false theorem"
    thm := cpu bl < 0 ∧ (cpu cl < 0) ⟹ (cpu al < 0).
    model := (¬ thm) model.
    "real counterexample"
    cl := model evaluate: cpu cl.
    bl := model evaluate: cpu bl.
    al := model evaluate: cpu al.
    cy := model evaluate: cpu carry.
    ov := model evaluate: cpu overflow.
    "check sum & flags"
    self
        assert: cl + bl equals: al;
        assert: cy equals: 1;
        assert: ov equals: 1
```

If a theorem is known to be true and its assertion fails, we can use the feature demonstrated in the listing above to find a counterexample and use it as a starting point for debugging the defective implementation.

### 7.7.4 Proofs

Although beyond the scope of this exposition, let's mention that we can ask Z3 to produce formal proofs of our theorems. In this way, Z3 becomes the *mathematician* that will give certainty about the theorems we happen to state in our tests, or that will exhibit a counterexample for us to debug.

## 7.8 Implications

The main insight of this work has been the realization that, while enhanced literal-injection techniques supported by DSLs offer a bridge between Smalltalk and other programming languages [cf. Helvetia, Renggli et al. (2010; 2009)], it is the combination with its other capabilities that expands the (already outstanding) expressiveness of the Smalltalk syntax to scientific notations used in Mathematics, Logic, Physics and presumably others. This is important because Smalltalk has shown to be inherently appropriate for capturing and producing scientific knowledge [cf. Notarfrancesco (2022), also Shingarov (2019) and Shingarov (2022)]. The implementation of self-hosted systems is facilitated by the possibilities enabled by Assembly [Section §5.3] and IL [Section §5.5]. Underlying this work is the realization of how much simpler it is to try extensions to the ANSI syntax when the Smalltalk compilation process is clearly structured and coded [Section §2].

The case study we have chosen illustrates the importance of having the possibility to extend the expressiveness of Smalltalk by increasing the fluency of binary selectors [Sections §3.2 and §3.3], incorporating Unicode symbols and characters in general to represent not just data but also message selectors [Section §1], and cradling foreign languages [Sections §3.1, §5 etc.].

Although non-trivial, our study was simple enough as to give visibility to several aspects that are relevant to programming and testing. Software development has become an *after the fact* activity, where practitioners are more inclined to reflect on the emergent behavior of the program as opposed to anticipating all the consequences derived from a thoughtful (static) analysis of the source code. This means that every effort put into leveraging the expression of (dynamic) logical relationships has the potential to extend the reach of inquire and reflection.

There is a paradigm shift from example-based testing to theorem formulation, formal proofs and automatic counterexample production. Our experiment also seems to indicate that theorem statement is not as far away from the standard developer idiosyncrasy as one might suspect at first glance. In fact, theorems are nothing but `Boolean` expressions on variables running on some domain. In our experience, well-equipped `IDE`s have the opportunity to extend the currently prevailing train of thought to new horizons. Under the light of this paradigm shift, example-based testing looks too modest compared to the advanced possibilities at our disposal, especially when the `IDE` elegantly galvanizes them as natural extensions of leading agile methodologies. Not just because of the robustness of the software but by the changes produced in our way of thinking.

### Acknowledgments

The authors want to express their gratitude to their copy-editor Diego Seguí, who carefully improved the English of the manuscript.

Over the years, many people have contributed to the design and implementation of Bee Smalltalk, notably Javier Pimás, Gerardo Richarte and Valeria Murgia. Other participants in this effort and long-time users of the system for the development of applications of industrial strength include Guillermo Amaral, Jean-Baptiste Arnaud, Alejandra de Bonis, Carlos E. Ferro, Sebastián Van Lacke, Adrián Somá and Jan Vrany.

Special thanks to Boris Shingarov and Jan Vrany for encouraging and supporting the authors in implementing an interface to the `Z3` library.

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

## A   Implementation Details

The source code used to implement our model of a `CPU` including each of its assembly instructions works with or without attachment to the `Z3` library [cf. Moura & Bjørner (2008)]. In other words, the code under testing remains unchanged regardless of whether it will run with numbers or symbols. This is an essential remark; otherwise we would not be testing actual source code, but an adapted version of it. As we already mentioned, that would pose an obstacle to the adoption of the testing methodology proposed by the framework.

For instance, here is the `Smalltalk` code for the `add` instruction:

```
add: operand1 _: operand2
    | op1 op2 add s n t u ov cy |
    op1 := operand1 valueOn: cpu.
    op2 := operand2 valueOn: cpu.
    add := op1 + op2.
    n := operand1 length.
    s := op1 bitAt: n.
    t := op2 bitAt: n.
    u := add bitAt: n.
    ov := s ≢ 0 ∧ (t ≢ 0) ∧ (u ≡ 0) ∨ (s ≡ 0 ∧ (t ≡ 0) ∧ (u ≡ 1)).
    cy := s ≡ 1 ∧ (t ≡ 1) ∧ (u ≡ 1) ∨ (s ≡ 1 ∧ (t ≡ 1) ∧ (u ≡ 0))
       ∨ (s ≡ 0 ∧ (u ≡ 0) ∧ (t ≡ 1)).
    cpu flags
        carry: cy;
        overflow: ov;
        sign: u ≡ 1;
        zero: add ≡ 0.
    ^add
```

Since `Z3` arithmetic aims at being general, it is not surprising that we have had to add special code for the behavior of the `flags` register of the `Intel` cpu. The `sub` instruction is similar. Many others are much simpler because they can be deferred to `Z3`, for example:

```
and: operand1 _: operand2
    | op1 op2 and s |
    op1 := operand1 valueOn: cpu.
    op2 := operand2 valueOn: cpu.
    and := op1 ∧ op2.
    s := and bitAt: operand1 length.
    cpu flags
        carry: 0;
        overflow: 0;
        sign: s ≡ 1;
        zero: and ≡ 0.
    ^and
```

## A.1 Execution

The methods involved in the symbolic execution of assembly code are:

```
assemble: aByteArray
    ram dump: aByteArray at: ip
```

```
step: n
    n timesRepeat: [
        decoder offset: ip value - 1.
        self execute: decoder next]
```

The former dumps the program on the **cpu**'s **ram** at the position indicated by the instruction pointer register **ip** in an unsophisticated copy operation. The latter brings us to the execution method, where the developer expresses the semantics of the instructions used by the `runtime`.

The canonical solution at this point consists in using the visitor pattern to produce the double dispatching needed to take care of the particularities of each instruction. Another alternative, the one we chose, is to abbreviate the intermediate indirection by means of a well-known metaprogramming technique that consists in building the selector and then performing it:

```
execute: instruction
    | selector n |
    selector := instruction mnemonic.
    n := instruction arity.
    n > 0 ifTrue: [
        selector := selector copyWith: $:.
        n - 1 timesRepeat: [selector := selector , '_:']].
    self
        moveIp;
        perform: selector asSymbol withArguments: instruction operands
```

For instance, the execution of a `mov` instruction would end up sending the message `mov:_:` to the **cpu**. Similarly, `add` will send `add:_:` etc. Thus, in order to support an instruction, the framework requires the implementation of the corresponding methods `mov:_:`, `add:_:` etc.

## A.2 Limits

As the reader might presume, not everything can be formally stated and symbolically proven. To the limitations that belong in the `Z3` library one must add those derived from the target language. In the case of `Smalltalk` there is an optimization that collides with one of the functions provided by `Z3`, namely, the inlining of the `ifTrue:ifFalse:` family of selectors. As is well known, most dialects chose to save cpu cycles by avoiding the creation of closures that would otherwise result from the arguments of these branching methods. Instead they *inline* the code occurring in the blocks and use `jump` bytecodes to select the fragment that must be executed. For instance, the method

```
max: aMagnitude
    ^self > aMagnitude ifTrue: [self] ifFalse: [aMagnitude]
```

which requires the receiver to be an instance of the class `Boolean`. Otherwise, the branching fails and the `Smalltalk` process signals a `NotABoolean` exception. On the part of Z3 we have the FFI call `Z3_mk_ite` that implements an `if-then-else` branching. And since its first argument must be of `BoolSort` it creates a conflict with the `Smalltalk` optimization. Even though the `Smalltalk` optimization is optional, care must be taken because its removal may impact the performance of the entire system. Modifying the `vm` so that it can handle the non-Boolean case, much as `MessageNotUnderstood` does, is not trivial because by the time the clash happens the closures for reifying the message are not available.

From the partial solutions at hand, one that can be readily implemented consists in retrying the exception after deoptimizing the offending method. This is why the actual implementation of the `step:` method repeats the following message the given number of times:

```
step
    | instruction |
    decoder offset: ip value - 1.
    instruction := [decoder next] on: NotABoolean do: [:ex |
        ex method deoptimizeFor: ex receiver.
        ex retry].
    self execute: instruction
```

where deoptimizeFor: installs a deoptimized version of the method specific to the instance that received the branching message. Better ways to overcome this issue would allow us to take symbolic computation in `Smalltalk` substantially farther.

