# OpenReview forum: "On Possible Extensions to the Smalltalk Syntax"
_FAST.org.ar/2022/Workshop — FAST Smalltalk 2022_

### Official Review · Reviewer_Fttf · 2022-11-01
**Interesting syntax ideas, motivating examples, but maybe does too much at once**

**Rating:** 8
**Confidence:** 5

**Review:**

The paper presents possible syntax extensions for the Smalltalk syntax

- In 3-6 we have some smaller "tricks" to extend ST80 syntax
- Sections 7-10 explain the idea of using pragma syntax to add  sections of foreign languages
-  11-15 presents smaller examples for this feature
- 17 presents a larger use case that uses all the extensions

- Denoting embedded languages with a <tag> is nicely motivated with examples
- All syntax extensions are motivated for the single ideas, then with everything together with a non-trivial example

Points to improve

Section 3
- "whose only element is" Can you explain what this element is (in term of AST Nodes)?

Section 3
- it is not clear who calls #hasJsonFormat, how do I use this hook? Can I use it for other cases?

Structure
- The "flat" structure of the paper is odd (18 sections, with section 17 being the only one with a sub-structure and very long compared to the others)
- it makes it difficult to navigate the paper

- Would it not make sense to put things into section
-- 3-6 is about some smaller ideas
-- 7-10 explain one topic
-- 11-15 are examples for the pragma syntax

- It feels that the paper tries to discuss too many things at once, but I think that structure could guide that better

- Section 17 feels like a whole workshop paper in itself. Considering that the paper is quite long (24 pages), it might be an idea to focus just on one idea
- Section 18 feels more like a discussion section for the "mini paper" that is Section 17.

- I wonder of Section 17 could be simplified and not present the work described as a contribution, just use it to validate the syntax experiments?